# Advancements in the Application of Ribosomally Synthesized and Post-Translationally Modified Peptides (RiPPs)

**DOI:** 10.3390/biom14040479

**Published:** 2024-04-15

**Authors:** Sang-Woo Han, Hyung-Sik Won

**Affiliations:** 1Department of Biotechnology, Research Institute (RIBHS) and College of Biomedical & Health Science, Konkuk University, Chungju 27478, Chungbuk, Republic of Korea; swhan524@kku.ac.kr; 2BK21 Project Team, Department of Applied Life Science, Graduate School, Konkuk University, Chungju 27478, Chungbuk, Republic of Korea

**Keywords:** bioactive peptides, genetic engineering, heterologous expression, high-throughput screening, RiPPs, synthetic biology

## Abstract

Ribosomally synthesized and post-translationally modified peptides (RiPPs) represent a significant potential for novel therapeutic applications because of their bioactive properties, stability, and specificity. RiPPs are synthesized on ribosomes, followed by intricate post-translational modifications (PTMs), crucial for their diverse structures and functions. PTMs, such as cyclization, methylation, and proteolysis, play crucial roles in enhancing RiPP stability and bioactivity. Advances in synthetic biology and bioinformatics have significantly advanced the field, introducing new methods for RiPP production and engineering. These methods encompass strategies for heterologous expression, genetic refactoring, and exploiting the substrate tolerance of tailoring enzymes to create novel RiPP analogs with improved or entirely new functions. Furthermore, the introduction and implementation of cutting-edge screening methods, including mRNA display, surface display, and two-hybrid systems, have expedited the identification of RiPPs with significant pharmaceutical potential. This comprehensive review not only discusses the current advancements in RiPP research but also the promising opportunities that leveraging these bioactive peptides for therapeutic applications presents, illustrating the synergy between traditional biochemistry and contemporary synthetic biology and genetic engineering approaches.

## 1. Introduction

Bioactive peptides are a fascinating group of natural products with significant potential in pharmaceuticals and biotechnology. The potent biological activities of bioactive peptides, including antimicrobial, antiviral, and antitumor properties, make them prime candidates for drug development. Bioactive peptides are classified into the following two major groups based on biosynthetic pathways: (1) ribosomally synthesized and post-translationally modified peptides (RiPPs) and (2) non-ribosomal peptides (NRPs). RiPPs (e.g., lanthipeptides and lasso peptides) have unique biosynthetic pathways that combine ribosomal synthesis with highly diverse and complex post-translational modifications [1], while NRPs (e.g., penicillin and vancomycin) are assembled by non-ribosomal peptide synthetases independently of the ribosome [2].

RiPPs constitute a significant class of natural products found across all domains of life, from bacteria to humans. Because of their bioactive properties, stability, and specificity, RiPPs have gained attention from various industries. In the pharmaceutical sector, RiPPs are investigated for their potential as novel therapeutics including antibiotics [3], antivirals [4], and anticancer agents [5], many of which are undergoing clinical trials or are FDA-approved (Figure 1) [6]. In agriculture, they are considered for use as eco-friendly biopesticides [7], contributing to sustainable farming practices. The food industry employs RiPPs, such as nisin, as natural preservatives to combat spoilage and pathogenic bacteria, thereby extending product shelf life while ensuring safety [8]. The wide-ranging utility of RiPPs highlights their significant value across various industries.

RiPPs originate from precursor peptides that are synthesized on ribosomes from the corresponding mRNA. These precursors typically consist of a leader peptide (or in some cases, a follower peptide) guiding post-translational modifications (PTMs) and a core peptide undergoing PTMs. After ribosomal synthesis, the precursor peptides undergo a series of PTMs that are responsible for the remarkable diversity in the structure and function of RiPPs. The PTMs can include processes like cyclization, dehydration, methylation, and cleavage of leader peptides [9,10,11]. Leader peptides are particularly important for PTMs owing to their effects on the specificity and activity of tailoring enzymes, sometimes keeping RiPPs inactive until the modifications are completed [12]. PTMs play a pivotal role in optimizing the therapeutic efficacy of RiPPs through a variety of mechanisms. By introducing modifications that increase lipophilicity or facilitate membrane interactions, PTMs can significantly enhance cell permeability, thereby enabling RiPPs to effectively target intracellular pathways [13]. These modifications not only improve the stability of RiPPs in biological environments by conferring resistance to proteolytic degradation and stabilizing their structures but also induce significant changes in their three-dimensional conformation, which is crucial for their biological activities [14]. Moreover, PTMs can adjust the binding characteristics and affinity of RiPPs towards specific targets, enabling more effective interactions at lower concentrations and introducing new functional groups or biochemical properties [15]. This selective post-translational tailoring enhances the physicochemical and biological attributes of RiPPs, positioning them as versatile and potent candidates for various therapeutic applications by fine-tuning their pharmacological properties to address specific clinical needs effectively.

RiPPs can be easily predicted and engineered because of their direct genetic encoding. Notably, advancements in genome sequencing have played a crucial role in the identification and characterization of RiPPs. The ribosomal origin of these peptides allows for the prediction of their chemical structures from genomic data, facilitating genome-driven RiPP discovery. This characteristic renders RiPPs an attractive target for bioengineering and synthetic biology efforts aimed at producing novel bioactive compounds. Heterologous expression of RiPP gene clusters in hosts like *Escherichia coli* [12,16,17,18,19,20,21] and *Streptomyces* sp. [22,23] is essential for elucidating these peptides and generating novel derivatives. Recent advances in synthetic biology and bioinformatics have significantly impacted research on RiPPs, particularly in the discovery of novel RiPPs and their engineering. The integration of high-throughput genome sequencing with sophisticated bioinformatic algorithms has enabled the prediction of RiPP biosynthetic pathways directly from genetic material. For example, specialized tools such as AntiSMASH [24], PRISM [25], and RODEO [26] have been developed for mining and annotating RiPP biosynthetic gene clusters, leading to an accelerated identification of new RiPPs [27,28]. On the other hand, synthetic biology facilitates in vivo and in vitro synthesis and screening of RiPPs by heterologous expression under diverse generic circuits [29]. Furthermore, these advances have exploited the inherent promiscuity within RiPP biosynthetic systems to generate a diverse array of engineered compounds with enhanced bioactivities and stability [29]. However, translating these gene clusters into known chemical entities remains challenging because of the complex nature of the PTMs and enzyme–substrate interactions within the cell. The lack of understanding of RiPPs restricts our ability to predict the complexity of RiPP biosynthetic gene clusters (BGCs), comprising multiple genes encoding peptides and proteins necessary for the biosynthetic process, and the diversity of PTMs. Therefore, the structural analysis of RiPP requires a multifaceted approach employing tandem mass spectrometry (MS) to identify PTMs and Nuclear Magnetic Resonance (NMR) to elucidate structural details and dynamics upon binding [30,31]. However, structural characterization is often hindered by low isolation yields from natural sources. To address these challenges, recent strategies have included the activation of silent biosynthetic gene clusters [23,32,33,34,35], the refactoring of biosynthetic gene clusters [22,36,37,38], and in vitro reconstruction of biosynthetic pathways [39,40,41].

In this work, we provide a comprehensive analysis of RiPPs, highlighting recent developments of synthetic biological systems and their applications in the production and engineering of RiPPs (Table 1).

Our discussion begins with the synthesis of precursor peptides on cellular ribosomes, detailing the specific and highly controlled PTMs such as cyclization, methylation, hydroxylation, acylation, and proteolysis, which are critical for maturing these precursors into bioactive compounds. We further explore the role of these modifications in enhancing RiPP stability and bioactivity, illustrating the cellular machinery’s precision in generating these molecules. The advances in synthetic biology for RiPP production are also examined, including strategies for heterologous expression and genetic refactoring to produce novel RiPP analogs. We highlight the substrate tolerance of tailoring enzymes as a key factor in generating diverse RiPP analogs and discuss the importance of leader peptides in directing PTMs. Finally, we introduce the latest screening methods for identifying functional RiPPs, preparing readers to appreciate the depth of research and technological innovation in the field of RiPP biosynthesis and function.

## 2. Biosynthetic Pathways of RiPPs

The biosynthesis of RiPPs commences in the cellular ribosomes, where precursor peptides are synthesized based on genetic information. These precursors typically consist of the following two distinct regions: the leader peptide and the core peptide (Figure 2). The leader peptide, positioned at the N-terminus (or occasionally at the C-terminus as a follower peptide), plays a critical role within the cell by guiding the subsequent PTMs of the core peptide. This leader peptide–core peptide architecture is essential for the controlled and specific modifications that the core peptide undergoes. The cellular machinery recognizes these leader peptides as targets for a series of enzymatic transformations that eventually result in the mature RiPP.

Within the cell, the core peptide undergoes various PTMs, a critical process for the functional diversity of RiPPs. Common modifications include the addition or alteration of functional groups, cyclization, and the formation of unique bond structures, such as thioether linkages. These modifications are not only diverse but also highly specific, often occurring at precise locations within the core peptide. The cellular environment thus plays a critical role in ensuring the correct folding and processing of these peptides, which is essential for the biological activity of the final RiPP product.

Major PTMs of RiPPs include cyclization, methylation, hydroxylation, acylation, and proteolysis. During cyclization, amino acid side chains can bridge across the chain, creating rings within the peptide backbone and forming circular structures, as exemplified by lanthipeptides [55]. Radical *S*-adenosylmethionine (SAM) enzymes, for instance, establish covalent bonds between side chains within the backbone (e.g., lanthipeptides) or head-to-tail connections (e.g., lasso peptides), generating complex cyclic scaffolds. These rings not only enhance stability but also influence interactions with target molecules, affecting biological activity. On the other hand, methyltransferases append methyl groups (–CH_3_) to specific nitrogen or oxygen atoms, subtly altering the RiPP’s structure. Methylation affects properties such as p*K*_a_, membrane interactions, and stability, thereby tuning the RiPP’s interaction with its biological targets. *N*-methylation, for example, increases stability against protease [56], while *O*-methylation in lanthipeptides and lasso peptides can adjust binding affinity [57,58]. Also, P450 enzymes incorporate hydroxyl groups (–OH) onto specific carbon atoms within the RiPP scaffold. This precise modification can activate RiPPs by modifying solubility and stability, with hydroxylation playing a vital role in antimicrobial activity in families like lassomycin [59]. Lastly, acyltransferases attach various acyl groups, such as acetyl or propionyl, to specific side chains. In surfactin A, acetylation affects surface properties, enhancing interactions with membranes and contributing to potent surfactant activity [60].

The precursor peptide contains the sequences for proteases beside the core peptides. Proteases cleave at specific peptide bonds, releasing the core peptide region from the precursor peptide that acts as a protective form of the RiPP. This proteolysis event not only activates the RiPP but also influences its final structure, such as revealing the active site. For more detailed information about proteolytic events, we refer to a comprehensive review published recently [11]. On the other hand, research efforts focusing on the tailoring enzymes for PTMs have provided a biochemical understanding of RiPP biosynthesis and strategies for engineering RiPPs. For example, previous studies have demonstrated the flexibility of tailoring enzymes for substrates (i.e., core peptides) [38,61,62]. The significant substrate tolerance of tailoring enzymes has not only highlighted the natural diversification of RiPPs but also facilitated the engineering of novel RiPP analogs. By exploiting this substrate tolerance and using synthetic biology techniques to manipulate genes encoding RiPP precursors, RiPP analogs have been synthesized with desired biological functions, such as enhanced antibiotic potency [44] and increased stability [45,63].

## 3. Advances in Synthetic Biology for RiPP Expression and Production

Synthetic biology provides innovative solutions to overcome the challenges associated with expression, engineering, and screening. A key advancement is the development of strategies targeting multiple synthetic biology levels, including individual proteins, pathways, metabolic flux, and host optimization. This approach significantly enhances the feasibility and effectiveness of RiPP preparation by tailoring the host’s metabolic machinery to support RiPP biosynthesis. Moreover, synthetic biology enables the engineering of RiPPs by reconstituting precursor peptides, wherein different functional groups are added to the core peptides. This interchangeability of substrate elements is crucial to tailoring RiPPs both in vivo and in vitro, thereby expanding the chemical and functional space of RiPPs.

RiPPs are produced via two distinct approaches including (1) natural biosynthesis and (2) heterologous expression. Natural biosynthesis utilizes the organism’s inherent metabolic pathways to produce RiPPs. Producing RiPPs in their native environment ensures correct folding and PTMs essential for their biological activity, as well as the natural diversity of RiPP structures and bioactivities. However, natural biosynthesis is limited by scalability, variability in yield, purity, and gene silencing [64]. To address these issues, considerable research efforts have focused on heterologous expression in surrogate hosts like *E. coli* or *Streptomyces* sp. Heterologous expression allows researchers to circumvent the complexity and the limitations of native genetic systems, typically using model organisms whose genetic manipulation and scale-up are easier than source organisms. This approach, however, presents challenges such as the complexity of reconstituting native biosynthetic machinery in a heterologous host and stability issues with precursor peptides [18,19,20]. Additionally, PTMs may occur differently between the host and the native producer, potentially affecting the final RiPP structure and activity. Despite these challenges, heterologous expression (Figure 3) has offered opportunities for discovery and innovation beyond the capabilities of natural biosynthesis.

### 3.1. Genetic Manipulation for Heterologous Expression

To address the complexities inherent in the genetic systems of RiPPs, two innovative strategies stand out including heterologous expression and genetic refactoring. Heterologous expression is a promising approach for activating silent BGCs identified by bioinformatic tools but not expressed under laboratory conditions. This method involves transferring BGCs to more manageable host organisms, such as *E. coli*, enabling the activation of these silent BGCs often under a foreign promoter. However, the cloning method based on PCR amplification is impractical for large BGCs, especially because of the introduction of mutations during PCR amplification [65].

To minimize PCR errors, methods that join multiple DNA fragments, occasionally coupled with de novo DNA synthesis, have emerged as reliable alternatives (Figure 3A). For instance, Gibson assembly facilitates the simultaneous assembly of multiple overlapping DNA fragments in a single reaction by combining exonuclease, polymerase, and ligase activities [66]. Using assembly-based methods, Wuisan et al. cloned darobactin A BGCs from various *Photorhabdus khanii* substrains [35]. However, assembly methods face severe limitations due to length [67] and GC content [68]. Direct cloning can simplify the cloning process and reduce the effort required to obtain desired BGCs. This approach bypasses the construction of genomic libraries and captures BGCs directly from genomic DNA without PCR amplification, proceeding through homologous recombination.

RecET recombination, originally identified in the Rac prophage of *E. coli*, comprises two proteins including RecE exonuclease and RecT annealing protein [69]. These proteins facilitate homologous recombination, integrating linear DNA fragments into the chromosome or plasmids of *E. coli* (Figure 3B). Fu et al. developed cloning tools mediated by RecET, termed linear plus linear homologous recombination (LLHR) and linear plus circular homologous recombination (LCHR), which are mechanistically distinct from conventional recombineering mediated by λ Redαβ [70]. Exonuclease can enhance the performance of RecET recombination especially in cloning large genomic regions (>50 kb) [34]. Wang et al. described the exonuclease combined with RecET recombination (ExoCET), which entails associating two DNA molecules outside the cell through a combination of in vitro exonuclease treatment and annealing facilitated by RecET homologous recombination (Figure 3C). However, the direct application of RecET-based recombination is primarily in *E. coli*, as the system relies on specific interactions with its cellular machinery [69]. For application in other organisms, analogous systems or engineered versions of RecET adapted to the cellular environment of the host organism are necessary. For example, *Saccharomyces cerevisiae* possesses a natural homologous recombination system, leading to the development of transformation-associated recombination (TAR) [71]. In TAR, a vector containing two homology arms is linearized and co-transformed with genomic DNA harboring the BGCs of interest into *S. cerevisiae* (Figure 3D). The yeast’s recombination machinery facilitates the integration of DNA fragments into the vector. For instance, Santos-Aberturas et al. successfully captured 31.7 kb of the thiovarsolin BGC, comprising 25 genes from *Streptomyces varsoviensis*, utilizing TAR [23].

In contrast, the Cre-*lox* recombination system, originating from bacteriophage P1, offers a genetic engineering platform applicable across a broad spectrum of organisms beyond its origins [72]. This system consists of the Cre recombinase enzyme and *lox*P recognition sites, enabling the seamless integration of DNA based on the orientation and placement of *lox*P sites. Unlike organism-specific methods such as TAR [71] and RecET [69], which are tailored to yeast and *E. coli*, respectively, the simplicity and universality of the Cre/*lox*P mechanism allow for its application in a diverse range of microbial systems [33]. Recently, Enghiad et al. described CAPTURE (Cas12a-assisted precise targeted cloning using in vivo Cre-*lox* recombination), which combines the precision of CRISPR-Cas12a genome editing with the flexibility of the Cre-*lox* recombination system (Figure 3E) [32]. CAPTURE employs the Cas12a enzyme for DNA digestion, T4 DNA polymerase for DNA assembly, and Cre-*lox* recombination for in vivo circularization of DNA, addressing the challenges of conventional cloning such as high GC content and sequence repeats. The researchers successfully cloned 43 uncharacterized BGCs, including probable lanthipeptide and lasso peptide BGCs, within 3–4 days, with sizes up to 113 kb. In another study, CAPTURE was applied to clone BGCs for daptide, a novel class of RiPPs characterized by an unusual (*S*)-*N*_2_,*N*_2_-dimethyl-1,2-propanediamine-modified C-terminus. Ren et al. [22] cloned daptide BGCs from *Microbacterium paraoxydans* DSM 15019 and expressed them in *Streptomyces albus* J1074, a versatile host for natural product pathway expression. This approach enabled the production, isolation, and detailed characterization of these unique peptides. Through heterologous expression, the team identified and analyzed daptides, revealing their distinctive bioactivities, including hemolytic activity.

### 3.2. Refactoring

Approaches to cloning entire BGCs may face challenges because of complex regulatory mechanisms inherent within BGCs and compatibility issues with heterologous hosts. Refactoring BGCs could overcome this challenge by circumventing the natural regulatory network [29]. This process simplifies and optimizes BGCs by selecting and reorganizing essential genes into operons and introducing synthetic regulatory elements, thereby fostering modularity and simplification (Figure 3F). Codon randomization is often employed to eliminate unidentified regulatory elements and promote efficient translation [29]. Cao et al. utilized refactoring to enhance the heterologous production of a novel RiPP, imiditides, whose BGC comprises a precursor peptide of NmaA and a tailoring enzyme of NmaM [38]. The refactoring involved the co-expression of genes by placing His_6_-SUMO-*NmaA* and untagged NmaM on separate expression plasmids. This approach enabled the successful heterologous expression and the PTM of imiditides. For more complex BGCs, researchers have implemented a plug-and-play refactoring strategy, where each gene module is constructed, assembled into a single plasmid, and interchangeably used within clusters. Ren et al. applied plug-and-play refactoring to identify essential genes for daptide biosynthesis using the mpa BGC [22]. Each codon-optimized mpa gene (mpaABCDM) was subcloned onto helper plasmids and combined via Golden Gate assembly to construct different versions of biosynthetic pathways, demonstrating the essential function of mpaABCDM in dipeptide production. Although refactoring offers numerous advantages, it requires significant time to prepare modular fragments. Leveraging robotics can provide one promising solution. Ayikpoe et al. developed a high-throughput pathway refactoring platform based on DNA synthesis and robotic assembly using Type IIS restriction enzymes [37]. With the refactoring of 96 bacterial RiPP BGCs identified by the RODEO tool [26], they successfully isolated 30 peptides spanning six RiPP classes [lanthipeptide, lasso peptide, graspetide, glycocin, linear azol(in)e-containing peptide (LAP), and thioamitide], with three peptides exhibiting antibiotic activity against multidrug-resistant bacterial pathogens known as ESKAPE. This platform facilitated the rapid evaluation of uncharacterized BGCs through automated pathway refactoring and heterologous expression. BGCs can be reconstituted by substituting existing genes and elements with synthetic counterparts. In 2023, King et al. demonstrated the systematic mining of lanthipeptide and lasso peptide BGCs from 2229 human microbiome genomes to identify antimicrobial peptides [36]. To address the challenges presented by the diverse origins of BGCs, they engineered synthetic gene clusters by incorporating codon optimization, synthetic regulatory elements including ribosome binding sites and terminators, a SUMO tag for precursor peptide stabilization, and a His tag for peptide purification. The synthetic gene clusters also featured a TEV protease site, replacing natural leader cleavage sites for simplified cleavage. Among seventy BGCs identified by the antiSMASH tool [24], twenty-three peptides (19 lanthipeptides and four lasso peptides) were functionally characterized, leading to the discovery of several RiPPs exhibiting activity against multidrug-resistant pathogens, including three RiPPs effective against vancomycin-resistant *Enterococci*. These findings underscore the potential of refactoring and heterologous expression as a potent strategy for the discovery of novel bioactive compounds, making a significant contribution to antimicrobial research.

### 3.3. Compartmentation

One major challenge in the production of RiPPs is the potential cytotoxicity of mature RiPPs to heterologous hosts, which hinders the discovery of novel RiPPs and the development of high-yield production systems. To mitigate this cytotoxicity, research efforts have focused on exploiting natural resistance mechanisms, such as efflux systems including peptide translocation [73,74] and transporters [75,76,77]. A notable study employed a compartmentation strategy for the expression of RiPPs, particularly focusing on lanthipeptides, through a synthetic biology approach in *E. coli* (Figure 3G) [12]. This strategy is crucial for overcoming the challenge of leader peptide removal—a bottleneck in heterologous RiPP production—by temporally programming leader peptide cleavage through protease compartmentalization and inducible cell autolysis. Specifically, it involves expressing the precursor peptide and biosynthetic enzymes in the cytosol, while compartmentalizing the protease to the periplasmic space to avoid premature interaction and potential cytotoxicity. Autolysis is induced using a temperature-controlled lysis gene cassette from bacteriophage λ, enabling the release of bioactive peptides after PTMs have been completed in the cytosol. Remarkably, this method simplifies the RiPP production process by facilitating in vivo leader peptide removal, significantly improving the throughput for discovering, characterizing, and engineering RiPPs. It also demonstrates the system’s effectiveness in producing bioactive lanthipeptides, such as haloduracin and lacticin 481, highlighting the method’s potential scalability and applicability to other RiPP classes, thereby revolutionizing RiPP engineering and discovery efforts.

### 3.4. Fusion Tags

The diversity of maturation mechanisms across different RiPPs presents a challenge in developing a universally applicable production system. For instance, transporters are indispensable for the maturation of certain RiPPs [78], but not essential for lasso peptide maturation [79]. A strategy that is applicable to a wide range of RiPPs involves the stabilization of RiPPs through the attachment of an additional tag to the N- or C-terminus of the precursor peptide (Figure 3H). One widely utilized tag is a small ubiquitin-like modifier (SUMO), which, when fused to proteins, beneficially influences their expression, folding, and solubility [16,17]. In 2022, Glassey et al. described the broad applicability of the SUMO fusion to 11 RiPP classes originating from diverse species [18]. They aimed to overcome the inherent challenges of peptide instability and functional expression in heterologous hosts such as *E. coli*. By fusing a SUMO tag to either the N- or C-terminus of the precursor peptides, this strategy stabilizes the expression of a broad spectrum of RiPPs, with the SUMO tag being proteolytically removed after PTMs. Remarkably, they successfully expressed 24 functional peptides out of 50 tested in *E. coli*, facilitating high-throughput screening and discovery of diverse RiPPs predicted by bioinformatic tools. Indeed, recent studies have exploited the SUMO fusion strategy for genome mining of RiPPs [36,38].

Fusion with fluorescent proteins, which are attractive partners for enhancing the solubility of recombinant proteins [80], is also applicable to RiPP expression. In a study by Vermeulen et al., plantaricin 423 and mundticin ST4SA, when fused with GFP at their N-terminus, were expressed in soluble forms in the host *E. coli* [19]. Additionally, Van Zyl et al. utilized mCherry for the heterologous expression of lanthipeptides such as nisin and clausin with N-terminal fusion [20]. Compared with other strategies, fluorescent tags enable the evaluation of RiPP expression levels through real-time fluorescence monitoring. For instance, the fluorescent intensity of GFP-MunX was compared under various conditions (e.g., IPTG concentration, expression time, and temperature) to optimize expression conditions, resulting in a yield of 12.4 mg of mundticin per liter of culture in *E. coli* [19].

### 3.5. Plasmid Copy Number

Very recently, Fernandez et al. highlighted the importance of selecting the appropriate plasmid vector and replicon, which can influence host cell viability and plasmid stability, for achieving high RiPP production yields [81]. Using capistruin—a lasso peptide—as a model system, the BGC was incorporated into different plasmids with varying replicons and then heterologously expressed in *Burkholderia* sp. FERM BP-3421. By increasing the plasmid copy number, they achieved a production yield of 240 mg/L, representing a 1.6-fold improvement over the previously optimized overproducer clone [42]. Interestingly, an increased plasmid copy number was associated with a prolonged lag phase during cell culture, indicating potential growth defects likely due to the antibiotic effect of the produced capistruin. This strategy was applied to the production of mycetolassin-15 and mycetolassin-18, novel lasso peptides originating from *Mycetohabitans* sp. B13. Contrary to capistruin, a higher plasmid copy number resulted in approximately a 2-fold reduction in production yield and a shorter lag phase, indicating that the effectiveness of the production system depends on the type of RiPP. This approach provided insights into the role of plasmid copy number in balancing peptide production with host cell viability and growth.

## 4. Strategies and Innovations in RiPP Engineering

Protein engineering for RiPPs (Figure 4) holds significant promise for both fundamental research and practical applications, ranging from discovering new bioactive compounds with therapeutic potentials to understanding biological mechanisms. This process aids in elucidating cellular processes and disease mechanisms through the modification of RiPPs and plays a crucial role in combating antibiotic resistance by providing new antimicrobial agents. Engineering RiPPs enhances their specificity, activity, stability, and bioavailability, making them more effective as therapeutic agents with fewer side effects.

### 4.1. Core Peptides

One interesting feature of the RiPP biosynthetic system is the substrate tolerance of tailoring enzymes, which implies the versatility of these enzymes with various core peptides. Previous studies have reported that only several residues of core peptides are critical for tailoring enzymes to catalyze PTMs, independent of the types and numbers of other residues [40,43,82]. This feature suggests an engineering strategy of introducing mutations into tolerant sites to generate myriad RiPP analogs (Figure 4A) [21,43]. For instance, a radical SAM enzyme encoded by *PapB* from *Paenibacillus polymyxa*, which catalyzes thioether cross-links between Cys and acidic residues (i.e., Asp and Glu) across diverse sequences, can accept various core peptides containing a Cys-X_n_-Asp motif (n = 0~6), even with D-amino acids at cross-linking sites [43,83]. Such capability extends the enzyme’s utility beyond conventional substrate specificities, enabling peptide engineering by introducing diverse amino acids at tolerant sites and incorporating D-amino acids at either of the intolerant sites to alter their biological activities. The enzyme’s flexibility facilitates the incorporation of unconventional amino acids and the crafting of complex peptide architectures, which are typically challenging via standard synthetic routes. Leveraging this unique substrate tolerance, researchers prepared an analog of the FDA-approved therapeutic agent octreotide, whose disulfide bond is replaced with a Cys-Glu thioether linkage. Although the analog’s biological function has not yet been profiled, this strategy presents a prospect for developing new peptide-based therapeutics, potentially improving biological effectiveness, stability, or pharmacological attributes.

The substrate tolerance of tailoring enzymes facilitates extensive engineering through the mix-and-match of diverse tailoring enzymes and core peptides, which is unprecedented in nature (Figure 4B). A recent study by Nguyen et al. exemplified this combinatorial strategy for macrocyclization using diverse core peptide backbones [82]. The researchers combined the tailoring enzymes MprC (cyclodehydratase), MprD (flavin-dependent oxidase), and PatG (subtilisin-like protease) with the core peptides MprE2, MprE5, MprE10, and PatE from *Methylovulum psychrotolerans*, aiming to create novel macrocyclized proteusin analogs with unique structural features. MprC facilitates the cyclodehydration of serine and threonine residues, MprD oxidizes azoline-containing peptides to azole-containing peptides, and PatG guides head-to-tail macrocyclization by recognizing an AYD sequence at the C-terminus of peptides. This approach leveraged the enzymes’ substrate tolerance to engineer RiPPs, demonstrating their potential as versatile biotechnological tools for generating diverse natural product libraries.

Zhao and Kuipers produced novel macrocyclic lanthipeptides, named thanacin and ripcin, by substituting the core peptide region in nisin BGCs with those of the antimicrobial peptides thanatin and rip-thanatin, respectively [44]. The capability of tailoring enzymes to catalyze foreign core peptides raised questions about whether PTMs occur simultaneously when core peptides are concatenated. They prepared a hybrid precursor peptide including the core peptide regions for both nisin and ripcin, generating a series of novel peptides named ripcin B–G depending on the length of the ripcin core peptide (13–18 residues) (Figure 4C). These hybrid lanthipeptides exhibited enhanced antimicrobial activity against *Staphylococcus aureus* and tested Gram-negative pathogens compared with either nisin or ripcin alone. Ripcin B–G were notable for their resistance to the nisin resistance protein, making them particularly attractive for selective antimicrobial applications in complex microbial environments.

Guo et al. combined two strategies—mutagenesis and hybrid construction—to enhance antimicrobial efficacy and stability [45]. They first created hybrid peptides by fusing domains from various nisin variants—nisin A, cesin, and rombocin—to produce novel entities like nirocin A and cerocin A. These hybrids exhibited improved action against methicillin-resistant *Staphylococcus aureus*. Subsequently, mutagenesis was employed to increase the peptides’ proteolytic stability, resulting in the discovery of cerocin V, which showed minimal degradation by trypsin. Their study highlighted the potential of combining domain modifications and targeted mutations to manipulate biological and physicochemical properties, generating novel bioactive molecules with promising therapeutic potential.

### 4.2. Leader Peptides

The leader peptide plays a crucial role in directing PTMs, guiding tailoring enzymes, and controlling the maturation of RiPPs, suggesting another strategy to modify RiPP properties. By altering leader peptides, researchers can influence the efficiency and type of PTMs, thereby creating RiPP variants with unique structures and activities. Burkhart et al. hypothesized that by fusing two leader peptides to construct a single chimeric leader peptide, two tailoring enzymes would bind to their respective regions on the chimeric leader peptide, facilitating the combination of PTMs originating from different RiPPs (Figure 4D) [84]. Specifically, the researchers combined an azoline-forming cyclodehydratase (HcaD/F) with a lanthipeptide synthetase (NisB/C) as a feasibility test. This approach demonstrated the potential to create hybrid RiPP products with diverse structural features, laying the groundwork for a broadly applicable platform for combinatorial RiPP biosynthesis. One potential issue with the chimeric leader peptide strategy is the maximum combination of leader peptides. The fusion of leader peptides could reduce the PTM efficiency, even with the repetition of identical leader peptides [85]. Additionally, some tailoring enzymes could work with a leader peptide comprising only one amino acid [86], indicating the possible production of a mixture of RiPPs, including molecules undergoing undesired PTMs. Thus, the sophisticated preparation of RiPP analogs through the fusion of multiple leader peptides requires an extensive understanding of enzyme–substrate recognition.

The leader peptide exchange strategy described by Franz and Koehnke offers a “plug-and-play” solution that can circumvent the complexity of the biosynthetic system (Figure 4E) [46]. They leveraged sortase A, which catalyzes transpeptidation, to exchange a pre-existing leader peptide with another, allowing a precursor peptide to carry only one leader peptide at a time. Specifically, sortase A cleaves a peptide bond within a specific recognition sequence (LPXTG) in the leader peptide and then forms a new peptide bond between the core peptide and the N-terminus of another leader peptide carrying the sortase recognition sequence. They modified the MdnA core peptide sequentially with cyclodehydration and macrocyclization by LynD and MdnC, leading to the preparation of a heterocycle-containing graspetide. This proof-of-concept demonstrates the leader peptide exchange as a potent tool that can facilitate the synthesis of innovative compounds with broad biological activities.

### 4.3. Tailoring Enzymes

Engineering tailoring enzymes, which directly contribute to the structural and functional diversity of RiPPs through PTMs, is pivotal for advancing RiPP engineering and providing synthetic biology tools (Figure 4F). For instance, prenyltransferases, which catalyze the attachment of prenyl groups to acceptor molecules, could enhance the biological activities of RiPPs by altering molecule lipophilicity and facilitating interaction with cellular targets. However, the broader application of prenyltransferases in producing diverse compounds has been limited by their strict specificity for prenyl donors. To overcome these limitations, Estrada et al. focused on PirF, a Tyr prenyltransferase with C10 isoprene donor (geranyl pyrophosphate, GPP) specificity [47]. Intriguingly, PirF shares over 70% sequence identity with prenyltransferases (e.g., PagF) that are only active toward dimethylallyl pyrophosphate (C5 isoprene donor, DMAPP) and not GPP. Through structure determination, the researchers identified that Gly221 in PirF corresponds to Phe222 in PagF in three-dimensional structures, likely influencing the size and hydrophobicity of the active site. Indeed, substituting Phe222 with alanine or glycine in PagF shifted the substrate preference from a C5 to a C10 isoprene donor, allowing for the use of alternative prenyl donors and expanding the applications of the tailoring enzyme.

The regulatory mechanism of tailoring enzymes, often requiring cognate leader peptides to exhibit activity, could obstruct the applications of engineered enzymes. Interestingly, a previous study by Levengood et al. revealed that supplying a leader peptide apart from a cognate core peptide (in *trans*) activated the tailoring enzyme for PTMs (Figure 4G) [87], suggesting a feasible strategy to mitigate the regulatory system. However, the requirement for large amounts of synthetic peptide made in *trans* activation economically unattractive. An alternative strategy is the fusion of a tailoring enzyme with a leader peptide, where they are covalently bound by additional linker sequences, to constitutively activate the enzyme. Following Oman et al.’s demonstration of the feasibility of this in *cis* activation with lantibiotic synthetase (Figure 4G) [88], subsequent studies demonstrated the generality of in *cis* activation for various enzymes, such as ATP-grasp ligases [89,90] and ATP-dependent cyclodehydratases [91]. Very recently, Lacerna et al. proposed an approach where both the leader and the core peptide were covalently attached to the tailoring enzyme (Figure 4G) [48]. By integrating the leader and core peptides into the enzyme, this method increases reaction efficiency because of the substrate’s proximity to the catalytic site, thereby enhancing the specificity and fidelity of complex cyclization reactions. Moreover, this method simplifies the production and purification processes of cyclic peptides, offering a streamlined approach to their isolation.

### 4.4. Combinatorial Approach

The engineering strategies involving leader peptides, core peptides, and tailoring enzymes can be synergistically combined. A recent study by Sarkar et al. employed multifaceted approaches to produce a broad range of *N*-methylated peptides [40]. Initially, they designed an in vivo expression system wherein OphMA (omphalotin methyltransferase) was fused to the N-terminal of an artificial peptide comprising various core peptides and two recognition sequences for PatA (protease) and PCY1/PsnB (macrocyclase). During heterologous expression, *N*-methylation occurred on the core peptide autocatalytically by in *cis*-activated OphMA. Subsequently, further PTMs involving peptide cleavage and macrocyclization were introduced in vitro by sequentially adding purified PatA and PCY1 (or PsnB), resulting in diverse *N*-methylated peptides. One challenge identified in their study is the limited substrate tolerance of OphMA, in contrast to PatA and PCY1, which have broad substrate specificity. This underscores the importance of promiscuous tailoring enzymes in RiPP engineering.

## 5. High-Throughput Screening Methods

Recent research efforts have focused on genome mining and peptide engineering to discover novel functional RiPPs as pharmaceutical candidates, through various screening assays as follows: (1) protein binding assays for affinity and binding inhibition tests, (2) growth inhibition assays for antimicrobial activity tests, and (3) cellular assays for cytotoxicity tests. To expedite the process, researchers rely on primary screening conducted in a high-throughput manner. High-throughput screening methods not only enable the identification of RiPPs with desired properties from a vast pool of candidates but also facilitate the characterization of their maturation mechanisms. Key strategies in these efforts include surface display [92,93,94,95], two-hybrid systems [53], and mRNA display [39,49,50,96], allowing researchers to rapidly screen peptide libraries. These screening methods rely on protein–RiPP interactions as well as the zone of inhibition assay [21,54] to assess physiological functions (Figure 5).

### 5.1. Surface Display

Surface display presents peptides or proteins on the surface of host cells, such as bacteria, yeast, or phages, directly linking the phenotype with its genotype. By genetically fusing the protein of interest to a cell wall or an anchor protein, this technique facilitates easy screening and selection of RiPPs with high affinity and specificity toward particular targets. This is crucial in drug development and enables the detailed study of RiPP–target interactions to understand their mechanisms of action. Despite challenges such as the need for proper peptide folding, display, and host-specific PTMs, surface display remains a powerful method for the high-throughput screening and engineering of RiPPs. It offers versatile platforms like bacterial [92], yeast [95], and phage display [93,94,95] for RiPP screening. For instance, phage display fuses the peptide of interest to either the N-terminus or C-terminus of a coat protein, such as pIII or pVIII of the M13 bacteriophage, enabling the rapid screening of libraries for targets by exposing the peptide on the surface of bacteriophages (Figure 5A) [97]. Urban et al. implemented the Sec pathway-based phage display to select lanthipeptide libraries specific to urokinase plasminogen activator and streptavidin [94]. Interestingly, PTMs only occurred with fusion to the C-terminus of coat protein pIII, not with N-terminal fusion. This is likely because N-terminal fusion directed the lanthipeptides toward the periplasm on the inner membrane during phage display, making them inaccessible to tailoring enzymes because the Sec pathway translocated the peptide in an unfolded state [94]. Conversely, the lanthipeptide with C-terminal fusion to pIII faced toward the cytoplasm during the assembly of coat proteins, ensuring sufficient time for PTMs. However, peptide fusion to the C-terminus of coat proteins may display inactive RiPPs because maturation typically accompanies the cleavage of a leader peptide by proteases [11]. Hetrick et al. addressed this issue by exploiting the Tat pathway, where translocation is accomplished in the folded state, by fusing NisA, the nisin-encoding gene, to the N-terminus of pIII [95]. They succeeded in displaying mature nisin on bacteriophages with the treatment of NisP protease to detach the leader peptide from the displayed nisin.

### 5.2. mRNA Display

Surface display technologies, while powerful, are subject to several limitations as follows: restricted library sizes (~10^9^) [96], the avidity effect arising from displaying multiple copies of peptides [98], and in vivo biases during processes such as transformation and translocation [99]. In contrast, mRNA display, wherein the phenotype (i.e., peptide) is covalently connected to the genotype (i.e., mRNA) via a puromycin link, offers significant advantages by accommodating larger library sizes (~10^13^), facilitating display in a monomeric context, and employing a simple display scaffold (Figure 5B) [96]. For instance, Bowler et al. utilized mRNA display to screen peptide libraries (~5 × 10^11^) against two cancer targets including the calcium and integrin-binding protein CIB1 and the immune checkpoint protein B7-H3 [39]. In constructing the library, they utilized microbial transglutaminase, a versatile enzyme for lysine–glutamine cyclization, to generate diverse macrocyclic peptides, followed by trypsin treatment to distinguish between cyclized and non-cyclized substrates. Subsequently, they selected potent peptides through affinity selection against CIB1 and B7-H3, leading to the high-throughput discovery of specific inhibitors.

mRNA display is also instrumental in the field of post-translational enzymology. Fleming et al. applied mRNA display to study the interaction between the tailoring enzyme PaaA and approximately 34 million PaaP variants, wherein six specific sites from T6 to I11 were randomly mutated, during the biosynthesis of the antibiotic Pantocin A [49]. This technique enabled them to explore the tailoring enzyme’s substrate tolerance and the impact of various mutations on enzyme activity, enhancing their understanding of peptide–protein interactions and the synthesis of novel RiPPs. Recent advances in computer science have opened a new era to predict biological interaction. Recently, Vinogradov et al. combined mRNA display with deep learning to investigate the substrate fitness landscapes of Ser dehydratase and YcaO cyclodehydratase involved in lactazole A biosynthesis [50]. This innovative approach generates extensive datasets from mRNA display, which are then analyzed using deep learning algorithms to predict enzymatic substrate preferences. By integrating deep learning, this platform offers a more precise mapping of catalytic preferences of tailoring enzymes, elucidating the molecular basis of cellular processes.

### 5.3. Two-Hybrid System

The ribosomal synthesis of RiPPs makes the two-hybrid system amenable to high-throughput screening for target proteins. This system is instrumental in detecting peptide–protein interactions in vivo by fusing the peptide/protein of interest to separate domains of a transcription factor (Figure 5C) [100]. The interaction between the peptide and the protein reconstitutes a functional transcription factor that activates a reporter gene, leading to color development or growth on selective media. Applied to RiPPs, the two-hybrid system facilitates the identification of novel tailoring enzymes essential for PTMs and RiPP candidates for drug development. Yang et al. employed a lanthipeptide library screening in an *E. coli* host cell through a bacterial reverse two-hybrid system based on the chimeric operator and the repressor of a bacteriophage regulatory system [53]. To identify a lanthipeptide inhibiting the critical protein–protein interaction necessary for HIV budding, they designed two fusion proteins including 434-human TSG101 UEV and P22-HIV p6. The bacteriophage proteins 434 and P22 create a functional repressor complex that inhibits the expression of reporter genes HIS3 and Kan^R^, which is essential for cell survival on specific media. The repression depends on the p6 and UEV protein–protein interaction, where the binding of a lanthipeptide disrupts the p6-UEV interaction, conferring a growth advantage. Screening approximately 10^6^ libraries led to the identification of a potent inhibitor, XY3-3, heralding a new era in the discovery and development of novel therapeutic agents.

### 5.4. Intein-Based Genetic Circuit

On the other hand, King et al. leveraged a split intein system for in vivo detection of protein–protein interactions, addressing the challenge of identifying peptides that bind to “undruggable” targets without predefined binding sites [51]. Inteins are protein segments capable of excising themselves and ligating the remaining proteins into a new protein through protein splicing. By constructing two chimeric proteins comprising the σ factorN (N-terminal domain)-NpuN-bait and RiPP-NpuC (C-terminal domain)-σ factorC, they converted peptide–protein binding events into the transcription of reporter genes such as GFP and luciferase through a genetic circuit (Figure 5D). Utilizing *E. coli* as the host, this system tested 10^8^ RiPP variants simultaneously, significantly surpassing traditional methods in throughput and specificity. By using the SARS-CoV Spike receptor-binding domain as bait, this approach identified AMK-1057, a probable therapeutic against the SARS-CoV-2 virus, underscoring its potential as a powerful tool for drug discovery based on synthetic biology and offering a promising outlook for targeting proteins previously considered undruggable.

### 5.5. Next-Generation Sequencing

For the high-throughput screening of novel antibiotics to inhibit RNA polymerase, Thokkadam et al. utilized next-generation sequencing (NGS) to analyze lasso peptide ubonodin variants (Figure 5E) [52]. The screening process involved a library of cells, each producing a distinct ubonodin variant. Upon induction, cells harboring ubonodin variants that inhibited RNA polymerase (RNAP) would perish, while those with either inactive or immature variants would survive. The variants retaining RNAP inhibition activity were identified by sequencing the plasmids from the surviving cell library. To ensure accuracy, five stages of sequencing were performed as follows: the naive library, cloning transformation, screen transformation, pre-IPTG, and post-IPTG. PCR amplification with barcoded primers was used to achieve an over-representation of library samples, ensuring adequate coverage during sequencing on an Illumina MiSeq platform. The analysis focused on the relative frequencies of amino acid substitutions, with increases indicating a loss in RNAP inhibition activity. This method not only facilitated the discovery of potential antibiotics for treating infections caused by *Burkholderia cepacia* complex pathogens but also enabled a comprehensive structure–activity analysis of ubonodin variants.

### 5.6. Zone of Inhibition Assay

The antimicrobial feature of RiPPs can be screened using the zone of inhibition assay, where microbes grow only in regions devoid of antibiotics. A limitation of this method is its labor-intensive nature and low throughput. Schmitt et al. developed an innovative strategy called nanoFleming to screen antibiotic candidates through the growth inhibition of target bacteria (Figure 5F) [54]. This method miniaturizes and parallelizes Fleming’s inhibition zone assay into a high-throughput format to screen large libraries of lanthipeptide variants that inhibit the growth of pathogenic bacteria. For the assay, two types of cells were prepared including mCherry-producing candidate cells that also secreted pre-lanthipeptide variants and GFP-producing sensor cells. These were immobilized in a 500 µm/65 nL alginate hydrogel compartment with a soluble protease. When a secreted lanthipeptide variant exhibited antimicrobial activity, the growth of the sensor cell was inhibited, leading to a decrease in green fluorescence intensity. Using this assay, they identified 11 peptides effective against bacteria showing immunity or resistance to nisin. Focusing on growth inhibition as a measure of antimicrobial activity, the nanoFleming platform emerges as particularly valuable in the development of new therapies to combat antibiotic-resistant bacteria.

The zone of growth inhibition is also an attractive method for robotic screening. In 2022, Guo et al. developed a semi-automated workflow in which lanthipeptide variant libraries were robotically constructed, expressed, and screened [21]. This workflow included an antimicrobial screening step by the zone of growth inhibition using microtiter plates to ensure compatibility with robotic automation. Using this workflow, they constructed a library of 380 single-site and 1373 triple-site mutants of HalA1, resulting in one variant with enhanced antimicrobial activity. Despite a few limitations, such as the poor correlation between the zone of growth inhibition assay using cell lysates and the specific activity using purified peptides, this automated workflow exemplifies the integration of synthetic biology and automation for the rapid and high-throughput characterization of natural products.

## 6. Conclusions

In this comprehensive analysis, we explore the vibrant and multifaceted landscape of biotechnological innovation and potential therapeutic discovery presented by RiPPs. The intricate biosynthetic pathways highlight the biological significance and complexity of RiPPs. The various PTMs that RiPPs undergo not only exemplify the diversity and specificity inherent in biological systems but also underscore the delicate balance between structure and function, crucial for the peptides’ bioactivity.

Advancements in synthetic biology and genetic engineering techniques have significantly broadened the scope of RiPP production and modification, overcoming previous limitations and opening new avenues for exploration. By engineering precursor peptides, tailoring enzymes, and host organisms, scientists can produce RiPPs with enhanced properties or entirely novel functions.

The evolution of screening methods, from traditional assays to cutting-edge technologies like mRNA display and next-generation sequencing, enables researchers to efficiently screen through vast libraries of variants. These techniques expedite the discovery of promising candidates and facilitate a deeper understanding of the intricate relationships between peptide structure, function, and biosynthetic machinery. Leveraging high-throughput and precise methodologies, the field is poised to uncover RiPPs with unique and potent biological activities, marking a significant stride toward addressing the need for new antimicrobial agents and therapeutic peptides.

In conclusion, the study of RiPP biosynthesis, engineering, and screening exemplifies the power of synthetic biology in unlocking nature’s mysteries and highlights the potential of RiPPs as a rich source of innovative therapeutic agents. As research progresses, the integration of advanced genetic engineering strategies and high-throughput screening methods will undoubtedly continue to push the boundaries of what is possible, leading to the development of novel RiPP-based applications in medicine, agriculture, and biotechnology.

## Figures and Tables

**Figure 1 biomolecules-14-00479-f001:**
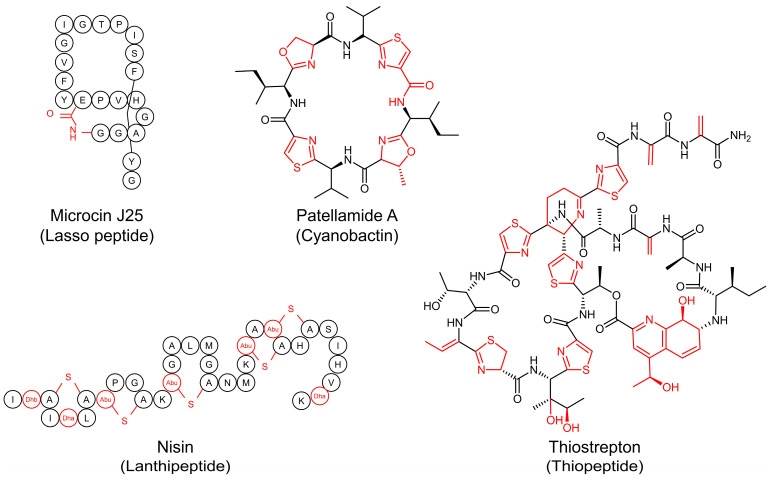
Representative therapeutic RiPPs and their classes. Letters in circles represent amino acids, and moieties undergoing modification are highlighted in red.

**Figure 2 biomolecules-14-00479-f002:**
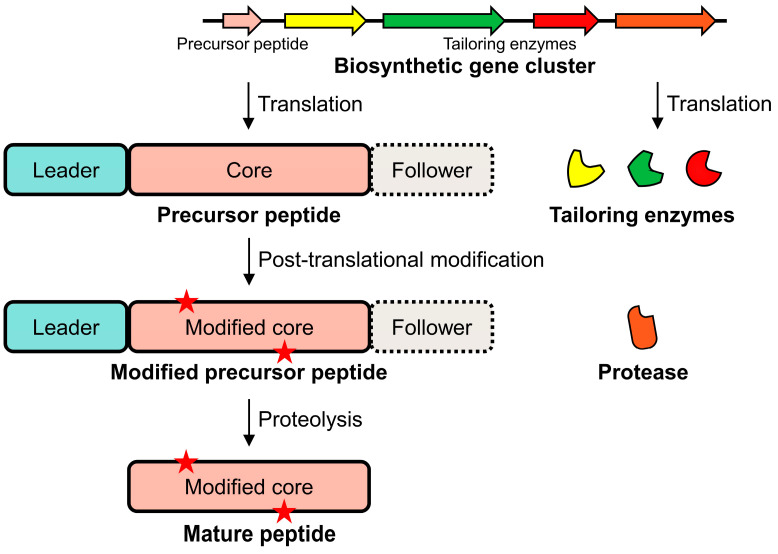
Schematic representation of the RiPP biosynthetic pathway. The biosynthetic gene cluster, which includes various genes responsible for RiPP synthesis, is translated into a precursor peptide and tailoring enzymes. After translation, tailoring enzymes, recruited by the leader (and/or follower) peptide, modify the core peptide. Subsequently, a protease cleaves the leader (and/or follower) peptide, resulting in the production of the mature peptide. Red stars represent the occurrence of PTM.

**Figure 3 biomolecules-14-00479-f003:**
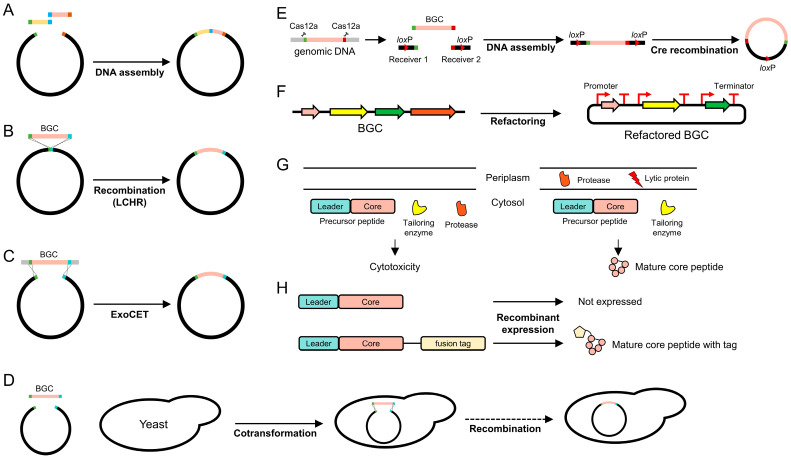
Genetic engineering for heterologous RiPP expression. (**A**) In DNA assembly methods such as Gibson assembly, multiple PCR products are ligated into an expression vector through a single isothermal reaction. (**B**) RecET facilitates homologous recombination between a lengthy linear fragment of a BGC and a vector (circular or linear) containing homologous regions. (**C**) ExoCET employs an exonuclease in addition to promoting recombination with longer fragments of BGC that carry non-homologous overhangs. (**D**) Upon transformation into yeast with fragments of BGC and a vector with homologous regions, the DNA fragments are assembled via the yeast’s native recombination system. (**E**) In the CAPTURE technique, the BGC fragment, isolated from genomic DNA by Cas12a, is ligated into synthetic receivers with *lox*P sites using DNA assembly methods, followed by circularization with the Cre enzyme. (**F**) Replacing native regulatory elements with uncharacterized mechanisms into well-understood systems facilitates the heterologous expression of selected BGC components, crucial for the biosynthesis of mature RiPP. (**G**) The expression of RiPP with bioactive properties triggers cell death in the host strain. However, by transporting a protease and a lytic protein to the periplasmic region and delaying the expression of the lytic protein, maturation of RiPP occurs, allowing the host cell to survive. (**H**) The addition of a fusion tag to a precursor peptide increases its stability and expression level, leading to an accumulation of mature RiPP in a heterologous cell.

**Figure 4 biomolecules-14-00479-f004:**
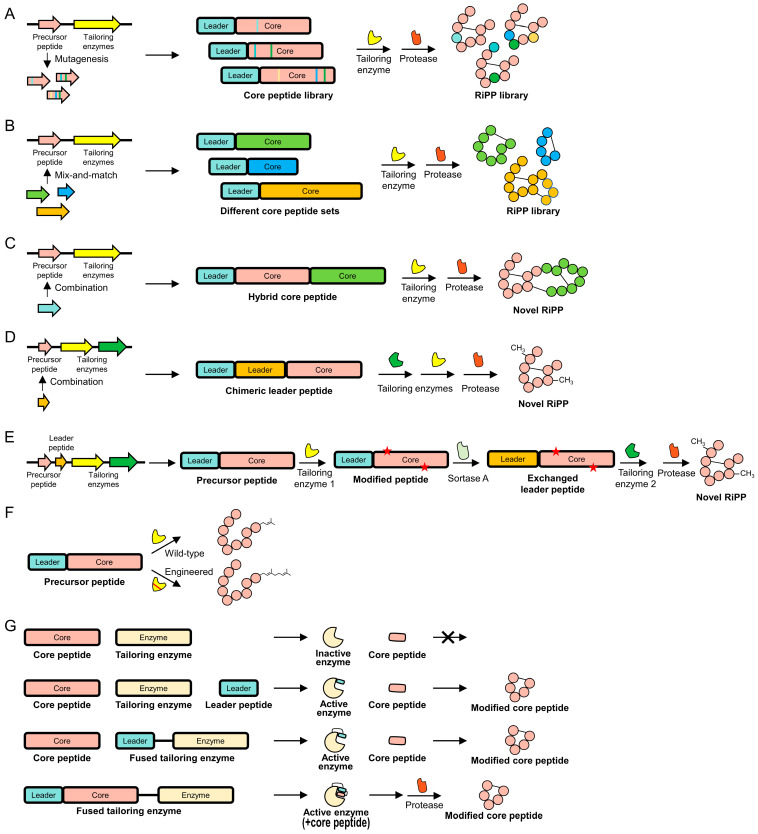
Strategies of RiPP engineering with respect to core peptides (**A**–**C**), leader peptides (**D**,**E**), and tailoring enzymes (**F**,**G**). (**A**–**C**) Addressing core peptides, substrate flexibilities of tailoring enzymes enable site-directed mutations (**A**), incorporation of foreign core peptides (**B**), and creation of hybrid core peptides with multiple domains (**C**), leading to a variety of RiPPs. (**D**,**E**) Utilizing leader peptides’ properties in guiding PTMs, diverse combinations of PTMs can be introduced on a single core peptide through chimeric leader peptides with multiple domains to guide PTMs (**D**) and leader peptide exchange using sortase A (**E**). (**F**) Protein engineering can enhance the substrate range of tailoring enzymes, broadening their application in generating RiPP variants. (**G**) The regulatory mechanism in tailoring enzyme activation can be simplified by introducing a free leader peptide and by fusing a tailoring enzyme with both a leader and a precursor peptide, simplifying the RiPP biosynthesis process.

**Figure 5 biomolecules-14-00479-f005:**
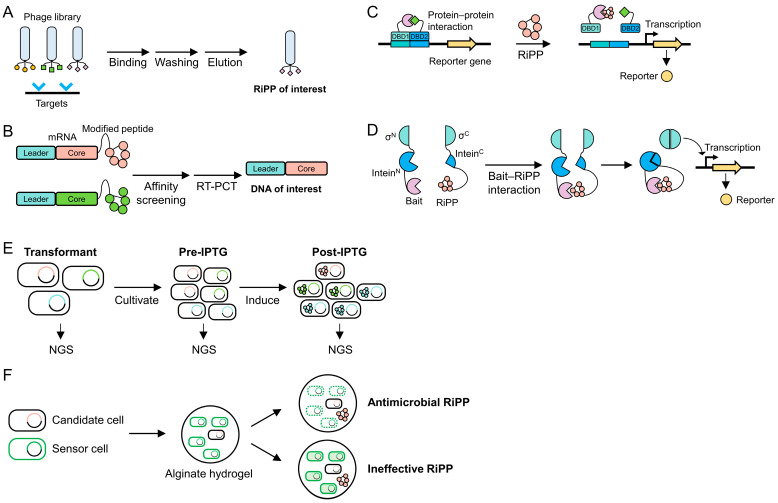
High-throughput screening methods utilized in RiPP research. (**A**) Phage display and (**B**) mRNA display techniques facilitate the straightforward detection of interactions between RiPPs and proteins or molecules. (**C**) The two-hybrid system can identify RiPPs that inhibit protein–protein interactions associated with diseases and infections. (**D**) Another in vivo screening method employs a genetic circuit based on intein, wherein the interaction between a RiPP and a target protein triggers the transcription of a reporter gene, allowing for the detection of RiPP–protein interactions. The antimicrobial activity of RiPPs can be assessed by their ability to inhibit the growth of (**E**) host cells and (**F**) neighboring cells. (**E**) Inhibition of host cell growth correlates with the concentration of RiPPs, as determined by NGS; a lower RiPP concentration signifies higher antimicrobial activity. (**F**) Inhibition of neighboring cell growth is evaluated using sensor cells that express a fluorescent protein; decreased fluorescence intensity indicates higher antimicrobial activity.

**Table 1 biomolecules-14-00479-t001:** A summary of the recently discovered or engineered RiPPs described in this review.

RiPP Product	Class	Biological Activity	Ref.
Thiovarsolin	Thioamitides	Unidentified	[23]
Daptide	Daptide	Hemolytic activity	[22]
Imiditide	Imiditide	Unidentified	[38]
Mycetolassin	Lasso peptide	Unidentified	[42]
7 RiPPs	Lanthipeptide, lasso peptide, LAP	Unidentified	[32]
30 RiPPs	Lanthipeptide, lasso peptide, graspetide, glycocin, LAP, thioamitide	Antimicrobial activity against ESKAPE pathogens	[37]
24 RiPPs	Lanthipeptide, lasso peptide	Antimicrobial activity against human pathogens	[36]
Octreotide analogs	Ranthipeptide	Unidentified	[43]
Hybrid RiPPs	Lanthipeptide	Antimicrobial activity against antibiotic-resistant MRSA strain	[44]
Hybrid RiPPs	Lanthipeptide	Antimicrobial activity against antibiotic-resistant MRSA strain	[45]
Hybrid RiPPs	Cyanobactin, microviridin	Unidentified	[46]
Prenylated lanthipeptides	Lanthipeptide	Unidentified	[47]
Cycle peptides	Cyanobactin	Unidentified	[48]
Cycle peptides	Cyanobactin	Unidentified	[40]
Pantocin A analogs	Pantocin	Unidentified	[49]
Lactazole analogs	Thiopeptide	Unidentified	[50]
Freyrasin analogs	Ranthipeptide	Binding to the SARS-CoV-2 Spike receptor	[51]
Ubonodin analogs	Lasso peptide	Antimicrobial activity against opportunistic human pathogens	[52]
Cycle peptides	Lasso peptide	Anticancer activity	[39]
XY3-3	Lanthipeptide	Inhibition to HIV infection	[53]
Hybrid RiPPs	Lanthipeptide	Antimicrobial activity against pathogenic bacteria	[54]
Halα analogs	Lanthipeptide	Antimicrobial activity	[21]

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
