# Peer review of "Advancements in the Application of Ribosomally Synthesized and Post-Translationally Modified Peptides (RiPPs)"

_biomolecules, 2024, doi:10.3390/biom14040479_

Round 1

Reviewer 1 Report

Comments and Suggestions for Authors

The authors of the manuscript 'Advancements in Ribosomally Synthesized and Post-Translationally Modified Peptides (RiPPs): Engineering, Production, and Screening for Therapeutic Applications' describe the many ways of producing and screening for effective RiPPs.

The manuscript is very interesting, well written and easy to read.

Some remarks:

The title is quite long and not 'catching'.
1) It would be nice to see a list (table) of nowadays used RiPPs in medicine.
2) It would help to show some structures of the used and general RiPPs, as these are the main subject, but not shown at all.
3) Although many techniques of producing these RiPPs are described, not much is written about their analysis, like activity measurements and structure analysis.

Reviewer 2 Report

Comments and Suggestions for Authors

The content and extension of this review exceeds my scope of knowledge, however I dare to recommend its publication.

There are at least 50 review papers published in the last 3 years, and it is difficult to ensure its differential contribution but the paper is well organized and helps to understand the relevance of Ribosomally synthesized and post-translationally modified peptides.

I recommend  the editor to make a decission taking in account those considerations.

Reviewer 3 Report

Comments and Suggestions for Authors

The manuscript is interesting. It presents a summary of the most relevant aspects concerning the RiPPs research issue. Peptides of the RiPPS class are an extremely interesting group of peptides, however, much less common than the 'classical' peptides. This may make the readership somewhat narrow. I did not notice any shortcomings of the manuscript. In my opinion, the issues concerning RiPPs are presented in a clear manner,. The graphics are of a very good standard. The only small detail for improvement that I noticed was the wording: 

l 63 three-dimensional configuration. The term configuration is usually used in a different context. A better option would be to use the term conformation (or possibly structure, although for most peptides I would recommend the former)

I would recommend the manuscript for publication in its current form.
